# Universal healthcare coverage and health service delivery before and during the COVID-19 pandemic: A difference-in-difference study of childhood immunization coverage from 195 countries

**Sooyoung Kim**[1], **Tyler Y. Headley**[2], **Yesim Tozan**[1]*

1 School of Global Public Health, New York University, New York, New York, United States of America,
2 New York University Abu Dhabi, Abu Dhabi, United Arab Emirates

* tozan@nyu.edu

## Abstract

**Data Availability Statement:** Original data can be accessed through following sources: • Global Health Security Index (GHSI) 2019: Hopkins U.

### Background

Several studies have indicated that universal health coverage (UHC) improves health service utilization and outcomes in countries. These studies, however, have primarily assessed UHC's peacetime impact, limiting our understanding of UHC's potential protective effects during public health crises such as the Coronavirus Disease 2019 (COVID-19) pandemic. We empirically explored whether countries' progress toward UHC is associated with differential COVID-19 impacts on childhood immunization coverage.

### Methods and findings

Using a quasi-experimental difference-in-difference (DiD) methodology, we quantified the relationship between UHC and childhood immunization coverage before and during the COVID-19 pandemic. The analysis considered 195 World Health Organization (WHO) member states and their ability to provision 12 out of 14 childhood vaccines between 2010 and 2020 as an outcome. We used the 2019 UHC Service Coverage Index (UHC SCI) to divide countries into a "high UHC index" group (UHC SCI $\geq$80) and the rest. All analyses included potential confounders including the calendar year, countries' income group per the World Bank classification, countries' geographical region as defined by WHO, and countries' preparedness for an epidemic/pandemic as represented by the Global Health Security Index 2019. For robustness, we replicated the analysis using a lower cutoff value of 50 for the UHC index. A total of 20,230 country-year observations were included in the study. The DiD estimators indicated that countries with a high UHC index (UHC SCI $\geq$80, $n = 35$) had a 2.70% smaller reduction in childhood immunization coverage during the pandemic year of 2020 as compared to the countries with UHC index less than 80 (DiD coefficient 2.70; 95% CI: 0.75, 4.65; $p$-value = 0.007). This relationship, however, became statistically nonsignificant at the lower cutoff value of UHC SCI <50 ($n = 60$). The study's primary limitation was

Global Health Security (GHS) Index. Retrieved from: https://www.ghsindex.org/wp-content/uploads/2019/10/2019-Global-Health-Security-Index.pdf. 2019. • Universal Healthcare Coverage Service Coverage Index (UHC SCI) 2019: Institute for Health Metrics and Evaluation. Global Burden of Disease Study 2019 (GBD 2019) UHC Effective Coverage Index 1990-2019. In:2020. (https://ghdx.healthdata.org/record/ihme-data/gbd-2019-uhc-effective-coverage-index-1990-2019) • WHO/UNICEF Joint Estimates of National Immunization Coverage: UNICEF. Immunization. In:2021. Retrieved from: https://data.unicef.org/topic/child-health/immunization/ • Country income groups categorization: World Bank. The World by Income and Region. https://datatopics.worldbank.org/world-development-indicators/the-world-by-income-and-region.html. Published 2021. Accessed September 14, 2021, 2021. All relevant data and analysis scripts uploaded in the github repository (https://github.com/sk9076/UHC_DID). The path to the repository is available in Supporting information files and is cited in the main manuscript.

**Funding:** The author(s) received no specific funding for this work.

**Competing interests:** The authors have declared that no competing interests exist.

**Abbreviations:** BCG, bacille Calmette–Guérin; COVID-19, Coronavirus Disease 2019; DiD, difference-in-difference; DTP, diphtheria, tetanus toxoid, and Pertussis; GHS, global health security; GHSI, Global Health Security Index; HEPB, hepatitis B vaccine; HIB, Haemophilus influenzae type B; HPV, human papillomavirus; IHME, Institute for Health Metrics and Evaluation; IPV, inactivated polio vaccine; IRB, institutional review board; MCV1, measles containing vaccine; PCV, pneumococcal conjugate vaccine; POL, polio containing vaccine; RCV, rubella containing vaccine; ROTAC, rotavirus vaccine; SD, standard deviation; SDG-3, Sustainable Development Goal 3; UHC, universal health coverage; UHC SCI, UHC Service Coverage Index; UNICEF, United Nations International Children's Emergency Fund; WHO, World Health Organization; YFV, yellow fever vaccine.

scarce data availability, which restricted our ability to account for confounders and to test our hypothesis for other relevant outcomes.

## Conclusions

We observed that countries with greater progress toward UHC were associated with significantly smaller declines in childhood immunization coverage during the pandemic. This identified association may potentially provide support for the importance of UHC in building health system resilience. Our findings strongly suggest that policymakers should continue to advocate for achieving UHC in coming years.

## Author summary

### Why was this study done?

- Studies to date have assessed the impact of public health crises on health systems almost exclusively during peacetime and/or under the framework of global health security (GHS).

- According to our literature review, we identified 15 articles that discussed the role of universal health coverage (UHC) on countries' health system performance in times of public health crises, none of which provided quantitative evidence to substantiate UHC's potential role in building health system resilience against external shocks like a pandemic.

- To our knowledge, our study is the first attempt to illustrate and quantify the association between countries' progress in UHC and countries' ability to protect essential health system delivery during the Coronavirus Disease 2019 (COVID-19) pandemic.

### What did the researchers do and find?

- We used a quasi-experimental difference-in-difference (DiD) study design and leveraged the COVID-19 pandemic as a natural experiment to quantify the effect of UHC on childhood immunization coverage before and during the pandemic.

- DiD estimators indicated that countries with a high UHC index (UHC SCI $\geq$80) were associated a 2.70% smaller decline in childhood immunization coverage during the pandemic year of 2020 as compared to the countries with UHC index less than 80 after adjusting for potential confounders.

### What do these findings mean?

- When combined with the extant empirical evidence, our findings strongly suggest that policymakers should continue to advocate for policies aimed at achieving UHC in coming years.

- This study also sets the stage for future research in understanding the synergistic impact of investments in GHS and UHC strategies on countries' health system resilience.

## Introduction

Achieving universal health coverage (UHC) is a pivotal target of the United Nations' Sustainable Development Goal 3 (SDG-3) [1]. Under UHC, it is envisioned that populations will have access to essential health services across the full spectrum of care, ranging from promotion to prevention, treatment, rehabilitation, and palliative care, without financial hardship [1]. Many studies to date have provided indications that UHC strategies improve health service coverage, utilization, and outcomes [2–6]. While the causal pathway remains contested, it is argued that UHC's emphasis on expanding pre-pooled funding mechanisms leads to a reduction in financial barriers to accessing necessary care and thereby results in improvements to population health [2]. Individual country studies further show that UHC helps to reduce inequalities in access to health services and increase utilization across sociodemographic groups, particularly for people with limited financial resources [3–6].

Despite this large evidence base, a robust quantitative assessment of the effects of UHC on health system performance and outcomes has proved challenging for 2 reasons [2]. First, disaggregated and standardized official data for public health, health system, and other pertinent indicators are scarce. Second, there exist many systems-wide contextual factors that may confound the relationship. Hence, the confluence of data and methodological constraints limit our ability to draw firm causal conclusions. For example, an observational study in Indonesia looked at a set of key health indicators, including maternal mortality ratio, infant mortality rate, and life expectancy, and found that UHC interventions achieved preliminary success in improving health equity and service access [7]. Another study from Indonesia used survival analysis to examine child cancer outcomes under UHC and reported significant improvements—especially among disadvantaged socioeconomic groups—in cancer survival and treatment failure [8]. However, the presence of multiple context-dependent confounders limited the generalizability of these findings to other similar healthcare settings.

Further, the extant research on UHC has largely focused on its peacetime impacts on health and health systems, which limits our understanding of UHC's potential contributions to countries' preparedness and response capacities during public health crises [9,10]. Studies examining the role of UHC in mitigating the health impacts of the Coronavirus Disease 2019 (COVID-19) pandemic are small in number and have generally focused on COVID-19 outcomes or the first wave of the pandemic in early 2020 [11–13]. Furthermore, given the paucity of empirical research in this area, UHC has generally not been considered integral to assessments of countries' preparedness and response capacities [9]. This is potentially a significant oversight given that countries' progress toward UHC requires not only overall health system strengthening but also sustainable pre-pooled funding mechanisms [2], both of which in theory should make countries more resilient to external shocks and more agile when responding to public health crises [10].

Establishing the causal effects of UHC on population health through an experimental study design is extremely challenging. However, an unforeseen public health crisis can serve as a natural experiment. The ongoing COVID-19 pandemic offers an opportunity to examine the role of UHC in safeguarding population health during a public health crisis. The COVID-19

pandemic itself needs little introduction; since early 2020, it has imposed severe burdens on countries' health systems, affecting the delivery of essential health services in varying but significant ways. With this in mind, we used a quasi-experimental difference-in-difference (DiD) design to compare differences in childhood immunization coverage based on countries' progress toward UHC, a proxy for countries' health system resilience, before and during the COVID-19 pandemic.

Immunization coverage, particularly the coverage of essential routine vaccines, is a good outcome measure to gauge the protective effects of UHC before, during, and after a crisis like the COVID-19 pandemic. First, immunization is considered an essential health service across all healthcare settings [14]. Second, vaccine coverage is an easily accessible and robust indicator at the global level; all countries have national immunization programs seeking to provide universal access to essential routine vaccines, especially for vulnerable populations, and report coverage data annually, which is further subject to verification processes for data quality [14]. Lastly, childhood immunizations—as represented by DTP-3 coverage among children and HPV vaccination coverage among teens—are a key input for the World Health Organization (WHO)'s UHC essential service coverage index and a specific target of SDG-3 because of the availability and ubiquity of data on childhood vaccine coverage [1]. The goal of this study was to examine the relationship between UHC and health service delivery during the COVID-19 pandemic shock. We hypothesized that greater progress toward UHC, represented by higher UHC Service Coverage Index (UHC SCI) 2019 values, would safeguard countries' ability to provide essential health services and minimize disruptions to service delivery during public health crises like the COVID-19 pandemic.

## Methods

### Data

National immunization data and trends were derived from the WHO/UNICEF Joint Estimates of National Immunization Coverage [15]. These data include annual vaccination coverage—in both absolute numbers and percentages—by country and type of vaccine. The dataset includes 195 countries and 14 childhood vaccines between 1997 and 2020. Details of data collection and calculation are described elsewhere [16,17]. For our analysis, we used a percentage coverage estimate, specifically the number of children who received a specific vaccine dose during a reported year (the numerator) divided by the number of children who were eligible to receive the vaccine during that year (the denominator). We merged the national immunization data with the UHC SCI 2019 obtained from the Institute for Health Metrics and Evaluation (IHME) [18]. UHC SCI 2019 is a robust and comparable measurement framework that measures countries' health system–effective coverage. Details of its framework and measurement are provided elsewhere [19]. In summary, the UHC index is a weighted aggregate of 23 indicators against 5 types of health services—promotion, prevention, treatment, rehabilitation, and palliation—and 5 age groups—newborn, children under 5 years, children and adolescents between 5 and 19 years, adults between 20 and 64 years, and older adults with age 65 years or more—across the life course. The index ranges from 0 to 100, with 100 indicating the highest effective service coverage. We used the World Bank's 2020 classification to assign each country to 1 of 4 income groups—high, upper-middle, lower-middle, or low income [20]. To control for countries' preparedness for an epidemic/pandemic, we incorporated data from the 2019 Global Health Security Index (GHSI) [21]. GHSI 2019 is an assessment of countries' health security and related capabilities necessary to prepare for future outbreaks including epidemics and pandemics. Details of its framework and measurement are provided elsewhere [22]. GHSI includes 37 indicators across 6 categories—prevention, detection and reporting, rapid

response, health system, compliance with international norms, and risk environment. The index ranges from 0 to 100, with 100 indicating the highest preparedness capabilities. The Supporting information presents the full list of countries included in the analysis (Table A in S1 File), as well as the compiled data from the aforementioned sources and the information on how to access the original data (S1 Text). Our study did not require institutional review board (IRB) approval as all of the data used in the study were publicly available and did not include any private, identifiable information.

## Data analysis

To test the study hypothesis, we adopted a quasi-experimental research design and conducted a DiD analysis [23]. DiD approaches are typically used to assess the causal effect of a policy or program by comparing the treatment group to a control group before and after an intervention, wherein a clear temporal cutoff pre- and post-intervention exists. DiD models have already been used to provide preliminary evidence of the effects of COVID-19 on several different health outcomes, including but not limited to neonatal outcomes [24], birth outcomes [25], or healthcare utilization rates [26].

DiD analyses need to satisfy 3 assumptions: first, a parallel pre-trend between the treatment and control groups prior to the intervention; second, no external spillovers of the outcome across the 2 groups; and third, that the intervention is unrelated to the outcome at baseline [23]. We demonstrated the parallel pre-trend in Fig 1 and presented empirical evidence of this parallel relationship in the Supporting information (Table B in S1 File). Further, a large body of literature demonstrates that one country's demand for essential vaccines (COVID-19 vaccines excluded) does not affect other countries' vaccine supplies due to the generally sufficient global manufacturing of essential vaccines [27]; rather, vaccine shortages are generally due to country-level logistics management, accessibility of health facilities, and health financing and policy issues [28]. Lastly, even in resource-limited countries with poor health service provisioning and lower UHC index values, childhood immunization programs are prioritized and perform relatively well compared to other service areas [1,14].

We divided countries into 2 groups based on their UHC index value, using a cutoff of 80. Countries with high UHC index values, defined as UHC SCI 2019 $\geq$80, were assigned to the treatment group, while the rest of the countries (UHC SCI 2019 <80) were assigned to the control group. The cutoff value of 80 was based on the published literature on this index, where UHC SCI $\geq$80 was operationalized to define the highest level of service coverage provision [19,29]. This was further confirmed through a visual inspection (Fig B in S1 File) of the distribution of the UHC SCI 2019 data, which showed that the data were roughly divided into 2 groups around this cutoff value. For robustness checks, we tested the alternative assumption that countries that made less progress toward UHC would be less able to manage disruptions to the delivery of essential health services during the pandemic. Accordingly, we used a lower cutoff value of 50 for the UHC index, which was close to the bottom quartile (UHC SCI 2019 <48.9) of the UHC SCI 2019 distribution, and assigned countries to the treatment group if UHC SCI 2019 <50 and to the control group if UHC SCI 2019 $\geq$50. The list of countries in their corresponding UHC SCI 2019 category is available in Table 1.

For the DiD analysis, we leveraged the COVID-19 pandemic to introduce a "prepost" variable wherein the years prior to 2020 (2010 to 2019) were defined as "pre" (0) and the pandemic year of 2020 was defined as "post" (1). While suboptimal for measuring the long-term treatment effects, DiD models only require one observation of data posttreatment to yield preliminary evidence of the treatment effect [23]. Unlike in a typical DiD design, the COVID-19 pandemic happened to countries on both ends of the UHC spectrum. While atypical, our

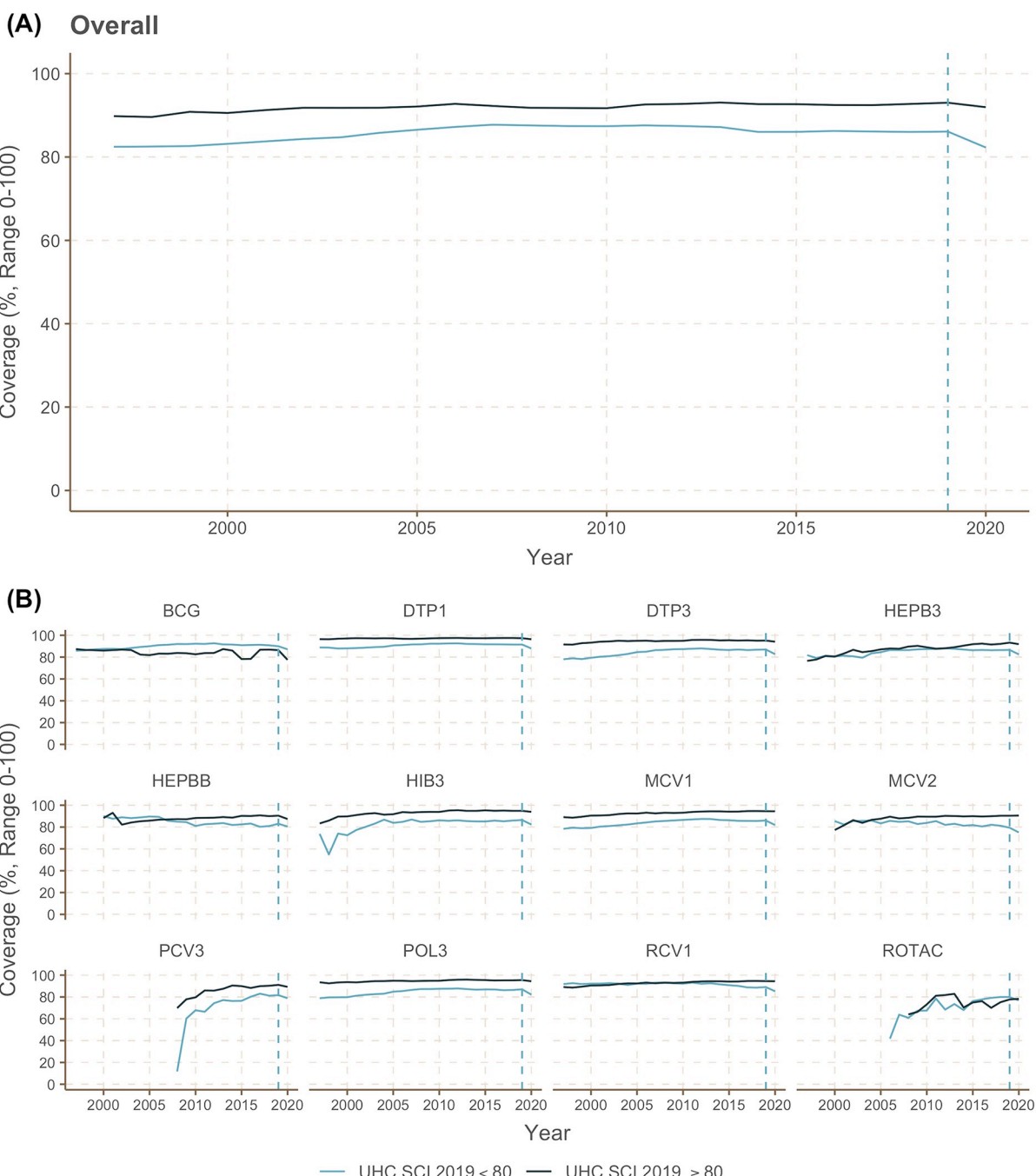

**Fig 1. Change in overall vaccine coverage (A) and vaccine-specific coverage (B) globally by UHC Service Coverage Index 2019 (UHC SCI ≥80 vs. UHC SCI <80) between 1997 and 2020.** Abbreviations: BCG = Bacille Calmette-Guérin; DTP1 = diphtheria, tetanus toxoid, and pertussis containing vaccine—first dose; DTP3 = diphtheria, tetanus toxoid, and pertussis containing vaccine—third dose; HEPB3 = third dose; HEPBB = hepatitis B vaccine—birth dose; HIB3 = Haemophilus influenzae type B containing vaccine; MCV1 = measles containing vaccine—first dose; MCV2 = measles containing vaccine—third dose; PCV3 = pneumococcal conjugate vaccine—third dose; POL3 = polio containing vaccine—third dose; RCV1 = rubella containing vaccine—first dose; ROTAC = rotavirus vaccine—second or third dose; UHC SCI = UHC Service Coverage Index.

**Table 1. Countries with UHC SCI 2019 ≥80 and UHC SCI 2019 <50.**

| | |
|---|---|
| **Countries with UHC SCI 2019 ≥80 (*n* = 35)** | Andorra, Australia, Austria, Belgium, Canada, Czechia, Denmark, Estonia, Finland, France, Germany, Greece, Iceland, Ireland, Israel, Italy, Japan, Kuwait, Luxembourg, Malta, Monaco, Netherlands, New Zealand, Norway, Portugal, Qatar, Republic of Korea, San Marino, Singapore, Slovenia, Spain, Sweden, Switzerland, United Kingdom, United States |
| **Countries with UHC SCI 2019 <80 and UHC SCI 2019 ≥50 (*n* = 100)** | Albania, Algeria, Antigua and Barbuda, Argentina, Armenia, Bahamas, Bahrain, Bangladesh, Barbados, Belarus, Belize, Bhutan, Bolivia, Bosnia and Herzegovina, Botswana, Brazil, Brunei, Bulgaria, Cabo Verde, Cambodia, Chile, China, Colombia, Cook Islands, Costa Rica, Croatia, Cuba, Cyprus, North Korea, Dominican Republic, Ecuador, Egypt, El Salvador, Eswatini, Gabon, Georgia, Grenada, Guatemala, Honduras, Hungary, Iran, Iraq, Jamaica, Jordan, Kazakhstan, Kenya, Kyrgyz Republic, Latvia, Lebanon, Libya, Lithuania, Malawi, Malaysia, Maldives, Mauritania, Mauritius, Mexico, Montenegro, Morocco, Namibia, Nicaragua, Oman, Palestine, Panama, Paraguay, Peru, Philippines, Poland, Moldova, North Macedonia, Romania, Russia, Rwanda, Saint Kitts and Nevis, Saint Lucia, Sao Tome and Principe, Saudi Arabia, Serbia, Seychelles, Slovakia, South Africa, Sri Lanka, Sudan, Suriname, Syria, Thailand, Tonga, Trinidad and Tobago, Tunisia, Turkey, Uganda, Ukraine, United Arab Emirates, Tanzania, Uruguay, Venezuela, Vietnam, Zambia, Zimbabwe |
| **Countries with UHC SCI 2019 <50 (*n* = 60)** | Afghanistan, Angola, Azerbaijan, Benin, Burkina Faso, Burundi, Cameroon, Central African Republic, Chad, Comoros, Congo, Cote d'Ivoire, Democratic Republic of the Congo, Djibouti, Equatorial Guinea, Eritrea, Ethiopia, Fiji, Gambia, Ghana, Guinea, Guinea Bissau, Guyana, Haiti, India, Indonesia, Kiribati, Lao People's Democratic Republic, Lesotho, Liberia, Madagascar, Mali, Marshall Islands, Federated States of Micronesia, Mongolia, Mozambique, Myanmar, Nepal, Niger, Nigeria, Niue, Pakistan, Palau, Papua New Guinea, Saint Vincent and the Grenadines, Samoa, Senegal, Sierra leone, Solomon Islands, Somalia, South Sudan, Tajikistan, Timor Leste, Togo, Turkmenistan, Tuvalu, Uzbekistan, Vanuatu, Yemen |

study still falls within the DiD framework given our assumption that health system resilience against COVID-19, the treatment effect of interest, was present at different levels within the 2 groups and was present prior to the COVID-19 pandemic.

The primary outcome was childhood immunization coverage, and the analysis was conducted for both overall vaccine coverage and specific vaccine coverage. We excluded the yellow fever vaccine (YFV) from the analysis because YFV is only administered in a limited number of high-risk countries, and the data are therefore not balanced between our stratified UHC categories. We also excluded the first dose of the inactivated polio vaccine (IPV-1) because IPV-1 was only introduced into the dataset after 2015, and the data were collected irregularly. After these exclusions, our analysis included 12 different childhood vaccines: bacille Calmette–Guérin (BCG); the first and third dose of diphtheria, tetanus toxoid, and pertussis containing vaccine (DTP1, DTP3); the birth dose of hepatitis B vaccine (HEPB-3); the third dose of hepatitis B containing vaccine (HEPBB); the third dose of *Haemophilus influenzae* type B containing vaccine (HIB3); the first and second doses of measles containing vaccine (MCV1, MCV2); the third dose of pneumococcal conjugate vaccine (PCV3); the third dose of polio containing vaccine (POL3); the second or third dose of rotavirus vaccine (ROTAC); and the first dose of rubella containing vaccine (RCV1).

We first used linear regression to estimate the base model without the DiD term (Eq 1) to illustrate the difference in immunization coverage between the pre- and post-COVID-19

period, as well as between the countries in the treatment and control groups. In the equation below, "Prepost" is the dummy variable that divides the time period into the pre-pandemic and pandemic periods, as explained above. "Treatment" is the dummy variable to indicate a country's assignment to either the treatment or control group.

$$(Immunization\ Coverage) = \beta_1 + \beta_2 * Prepost + \beta_3 * Treatment + Z + \epsilon$$

We then used linear regression to estimate a DiD model for the effect of UHC on childhood immunization coverage pre-pandemic and during the pandemic (Eq 2).

$$\begin{aligned}(Immunization\ Coverage)\\ = \beta_1 + \beta_2 * Prepost + \beta_3 * Treatment + \beta_4 * Prepost * Treatment + Z + \epsilon\end{aligned}$$

The DiD estimator is represented by an interaction term between the Prepost and Treatment dummy variables in a regression model, and the coefficient $\beta_4$ quantifies the causal impact of UHC on health system resilience after adjusting for all covariates. In all the analyses performed, we controlled for the following covariates as represented by the vector Z in the equation: calendar year, countries' income group as per the World Bank classification [20], geographical location based on countries' memberships to the WHO regional offices [30], and countries' preparedness for an epidemic/pandemic as represented by the GHSI [31].

We used linear regression because our outcome variable was continuous (range: 0 to 100) and because the pre-period trend in immunization coverage (2010 to 2019) appeared to be linear. Even if some countries' immunization coverage for certain vaccines were close to 100%, this should have not affected our analysis, which aimed to quantify the reduction in coverage, which brings coverage down from 100%. However, we limited our interpretation of the results to only the DiD coefficient ($\beta_4$) because the limitation of our linear regression methodology was that predicted values could be obtained beyond a realistic range (0 to 100).

The analysis was initially conceptualized and planned in September 2021, conducted between September 2021 and January 2022, and was revised based on peer review in April 2022. As part of the revisions, 2 additional analyses were conducted to ensure the robustness of findings. First, we tested whether shortening the pre-period to 2015 to 2019 would change the DiD estimate. Second, we replicated the same analysis using a sliding scale of cutoff values between 50 and 80 with step increments of 5 to evaluate the threshold of UHC SCI 2019 where significant resilience occurs. Throughout the analysis, all statistical tests are performed two-sided, and we use the threshold of 0.05 ($P < 0.05$) for stating the statistical significance. All analyses were conducted using R software (version 3.6.3), and the data and code used in the analysis are available in a public repository detailed in the Supporting information (S1 Text). This study is reported in accordance with the Strengthening the Reporting of Observational Studies in Epidemiology (STROBE) guidelines (S1 STROBE Checklist).

## Results

A total of 38,139 country-year observations were included in the collated data, of which 1,658 took place after the COVID-19 pandemic began. Fig 1 shows the mean childhood immunization coverage rate overall and by type of vaccine from 1997 to 2020. Prior to 2010, many countries, especially those with lower UHC index values, showed rapid improvements in immunization coverage each year; we thus performed the DiD analysis on the data from 2010 onward to fulfill the requisite parallel pre-trend assumption. This criterion resulted in a total sample size of 20,230, of which 1,658 (8.2%) observations took place during the pandemic.

Of the 18,572 observations over the pre-pandemic period, those countries with a UHC index greater than or equal to 80 ($N = 3,223$) had an average childhood immunization

**Table 2. Summary characteristics of the countries included in the analysis (*N* = 195) based on the cutoff value of 80 for the UHC SCI 2019.**

| | UHC Index <80 (*N* = 160) | UHC Index > = 80 (*N* = 35) | Overall (*N* = 195) |
|---|---|---|---|
| **UHC SCI[a] 2019** | | | |
| Mean (SD[b]) | 54.6 (11.9) | 88.0 (4.60) | 60.6 (16.8) |
| **World Bank Income group** | | | |
| High | 26 (16.3%) | 35 (100%) | 61 (31.3%) |
| Upper-Middle | 54 (33.8%) | 0 (0%) | 54 (27.7%) |
| Lower-Middle | 49 (30.6%) | 0 (0%) | 49 (25.1%) |
| Low | 30 (18.8%) | 0 (0%) | 30 (15.4%) |
| **WHO Region** | | | |
| Africa | 47 (29.4%) | 0 (0%) | 47 (24.1%) |
| Americas | 33 (20.6%) | 2 (5.7%) | 35 (17.9%) |
| Eastern Mediterranean | 20 (12.5%) | 2 (5.7%) | 22 (11.3%) |
| Europe | 27 (16.9%) | 26 (74.3%) | 53 (27.2%) |
| South East Asia | 11 (6.9%) | 0 (0%) | 11 (5.6%) |
| Western Pacific | 22 (13.8%) | 5 (14.3%) | 27 (13.8%) |
| **Global Health Security Index 2019** | | | |
| Mean (SD[b]) | 36.1 (11.2) | 58.7 (13.8) | 40.2 (14.6) |

[a]UHC SCI: Universal Health Coverage Service Coverage Index.

[b]Standard deviation.

coverage rate of 92.6% (standard deviation [SD] 9.8), whereas those with a UHC index less than 80 (*N* = 15,349) had an average immunization rate of 86.6% (SD 16.0). Of the 1,658 observations that took place during the pandemic year of 2020, countries with the high UHC index (UHC SCI 2019 ≥80) (*N* = 232) had an average childhood immunization coverage rate of 92.0% (SD 9.0), and countries with the UHC index lower than 80 (*N* = 1,426) had an average coverage rate of 82.3% (SD 16.0). Summaries of countries' characteristics in the treatment and control groups based on the cutoff values of 80 and 50 are provided in Table 2 and the Supporting information (Tables C in S1 File).

The results of the adjusted base model and the DiD model are shown in Table 3, and the unadjusted base model and the DiD model are provided in the Supporting information (Table A in S2 File). Across both the treatment and control groups, the global childhood immunization coverage rate in 2020 was 2.72% lower than the average of the period from 2010 to 2019 when all other covariates were held constant (base model coefficient for Prepost = −2.72%; 95% CI: −3.50, −1.95; *p*-value < 0.001). Using the DiD model, we found that countries with a high UHC index (UHC SCI 2019 ≥80) had a 2.70% smaller reduction in overall childhood immunization coverage during the pandemic year of 2020 (DiD model coefficient for Prepost * Treatment = 2.70; 95% CI: 0.75, 4.65; *p*-value = 0.007) as compared to the control group. In other words, when controlling for all other covariates, countries with high UHC index values experienced almost no decline in immunization coverage (DiD model combined effect size for Prepost and Prepost * Treatment = −0.41; 95% CI: −3.18, 2.36) as compared to a significant decline (DiD model coefficient for Prepost = −3.11; 95% CI: −3.93, −2.29; *p*-value < 0.001) in coverage among countries with low UHC index values (UHC SCI <80) (Fig 2).

As a robustness check, we first used a lower threshold of UHC SCI <50 to group countries into treatment and control groups (Supporting information Table Z in S2 File for unadjusted analysis, Table AA in S2 File for adjusted analysis). In this analysis, we found that the adjusted

**Table 3. DiD analysis results of countries with high UHC index values (UHC SCI ≥80) vs. all other countries (UHC SCI <80) in childhood immunization coverage in pre-pandemic period (2010–2019) compared with pandemic period (2020)—Adjusted for calendar year, pandemic preparedness, country income group, geographic region, and vaccine types.**

| Variable | Base model | | | DiD model | | |
|---|---|---|---|---|---|---|
| | Coefficient | 95% CI[a] | p-value | Coefficient | 95% CI[a] | p-value |
| Intercept | 47.49 | (−91.18, 186.16) | 0.502 | 48.22 | (−90.43, 186.87) | 0.495 |
| Year | 0.02 | (−0.05, 0.08) | 0.651 | 0.02 | (−0.05, 0.08) | 0.659 |
| GHSI[b] 2019 | 0.06 | (0.04, 0.08) | <0.001 | 0.06 | (0.04, 0.08) | <0.001 |
| World Bank Income Group (Reference category: Low) | | | | | | |
| Lower-middle | 7.11 | (6.44, 7.78) | <0.001 | 7.11 | (6.44, 7.78) | <0.001 |
| Upper-middle | 11.43 | (10.69, 12.17) | <0.001 | 11.42 | (10.68, 12.16) | <0.001 |
| High | 17.02 | (16.20, 17.85) | <0.001 | 17.01 | (16.19, 17.84) | <0.001 |
| WHO Region (Reference category: Americas) | | | | | | |
| Europe | 2.62 | (2.01, 3.23) | <0.001 | 2.61 | (2.01, 3.22) | <0.001 |
| Western Pacific | 0.01 | (−0.68, 0.69) | 0.986 | 0.01 | (−0.68, 0.69) | 0.981 |
| Eastern Mediterranean | −0.38 | (−1.11, 0.35) | 0.309 | −0.37 | (−1.10, 0.36) | 0.315 |
| Southeast Asia | 3.87 | (2.90, 4.83) | <0.001 | 3.87 | (2.91, 4.83) | <0.001 |
| Africa | −0.73 | (−1.43, −0.03) | 0.042 | −0.72 | (−1.43, −0.02) | 0.043 |
| Vaccine type (Reference category: BCG[o]) | | | | | | |
| DTP1[c] | 0.59 | (−0.28, 1.46) | 0.183 | 0.59 | (−0.28, 1.46) | 0.184 |
| DTP3[d] | −3.85 | (−4.72, −2.98) | <0.001 | −3.85 | (−4.72, −2.98) | <0.001 |
| HEPB3[e] | −4.64 | (−5.52, −3.76) | <0.001 | −4.64 | (−5.52, −3.76) | <0.001 |
| HEPBB[f] | −10.98 | (−12.11, −9.86) | <0.001 | −10.97 | (−12.09, −9.85) | <0.001 |
| HIB3[g] | −4.95 | (−5.83, −4.07) | <0.001 | −4.95 | (−5.83, −4.07) | <0.001 |
| MCV1[h] | −4.63 | (−5.50, −3.76) | <0.001 | −4.63 | (−5.50, −3.76) | <0.001 |
| MCV2[i] | −10.73 | (−11.66, −9.81) | <0.001 | −10.73 | (−11.66, −9.80) | <0.001 |
| PCV3[j] | −11.54 | (−12.56, −10.52) | <0.001 | −11.53 | (−12.55, −10.51) | <0.001 |
| POL3[k] | −3.93 | (−4.80, −3.06) | <0.001 | −3.93 | (−4.80, −3.06) | <0.001 |
| RCV1[l] | −3.22 | (−4.16, −2.28) | <0.001 | −3.22 | (−4.15, −2.28) | <0.001 |
| ROTAC[m] | −14.57 | (−15.74, −13.39) | <0.001 | −14.58 | (−15.75, −13.40) | <0.001 |
| DiD variables | | | | | | |
| Pre/Post | −2.72 | (−3.50, −1.95) | <0.001 | −3.11 | (−3.93, −2.29) | <0.001 |
| UHC SCI[n] 2019 ≥80 | −3.74 | (−4.50, −2.97) | <0.001 | −3.92 | (−4.69, −3.14) | <0.001 |
| Pre/Post * UHC[n] SCI ≥80 | | | | 2.7 | (0.75, 4.65) | 0.007 |

[a]Confidence interval.

[b]Global Health Security Index.

[c]Diphtheria, tetanus toxoid, and pertussis containing vaccine—first dose.

[d]Diphtheria, tetanus toxoid, and pertussis containing vaccine—third dose.

[e]Hepatitis B vaccine—third dose.

[f]Hepatitis B vaccine—birth dose.

[g]*Haemophilus influenzae* type B containing vaccine.

[h]Measles containing vaccine—first dose.

[i]Measles containing vaccine—third dose.

[j]Pneumococcal conjugate vaccine—third dose.

[k]Polio containing vaccine—third dose.

[l]Rubella containing vaccine—first dose.

[m]Rotavirus vaccine—second or third dose.

[n]UHC Service Coverage Index.

[o]Bacille Calmette–Guérin.

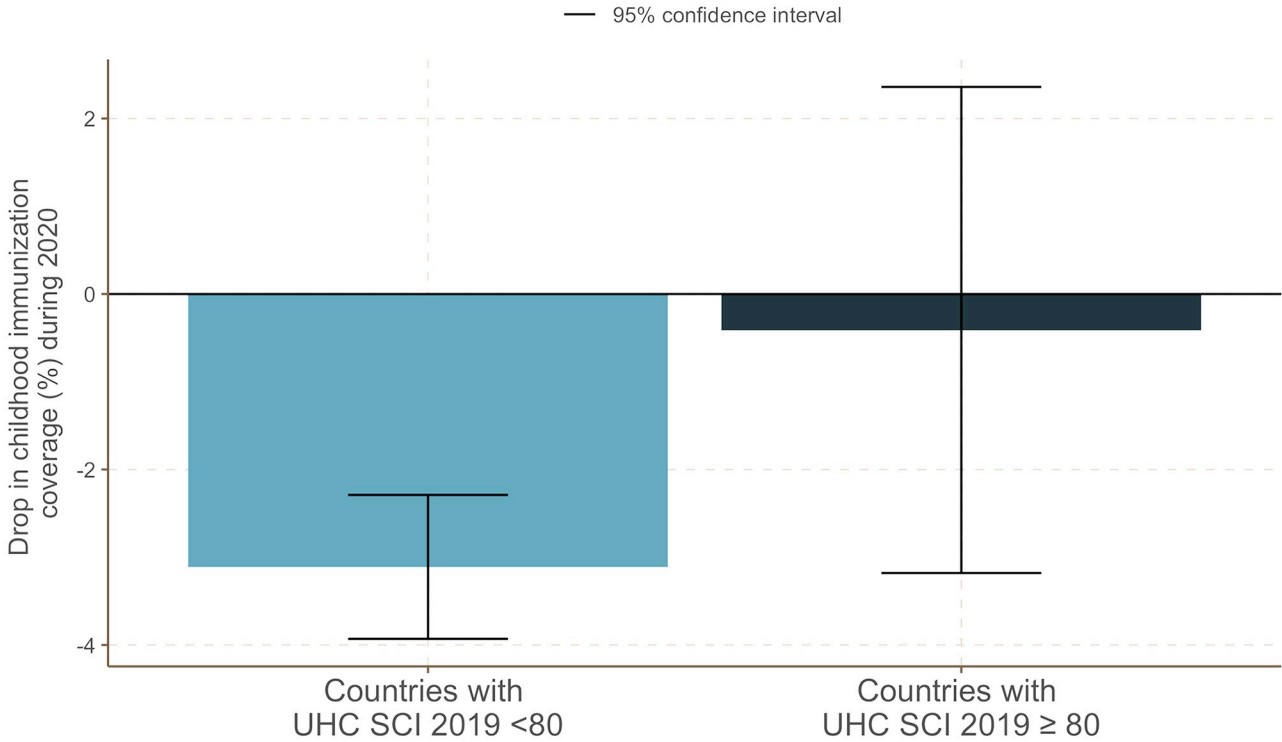

**Fig 2. Differential drop in childhood immunization coverage (%) during the COVID-19 pandemic between the countries with high UHC index (UHC SCI 2019 ≥80) and the rest (UHC SCI 2019 <80).** Abbreviation: UHC = Universal healthcare coverage; SCI = Service coverage index.

DiD coefficient was not significant (coefficient −0.56; 95% CI: −1.98, 0.86; *p*-value = 0.44), suggesting that there was no differential impact of a lower threshold of UHC on childhood immunization coverage during the pandemic. This finding may imply that achieving a certain level of UHC might be critical for the protective benefit of UHC to take effect. Subsequently, when we repeated the same analysis using the sliding threshold of UHC SCI between 50 and 80 with step increments of 5, we observed a significant association between UHC and a differential drop in immunization coverage during the pandemic from the threshold value of 65 and above (Supporting information Fig A and Table A in S3 File). Changing the pre-period from 2010 to 2019 to 2015 to 2019 did not change the overall findings (Supporting information Tables B–E in S3 File).

The DiD results using the original threshold for the UHC index (treatment: UHC SCI 2019 ≥80; control: UHC SCI 2019 <80) for each individual childhood vaccine are presented in Table 4, and full regression results are available in the Supporting information (Tables D–Y in S2 File). None but one (RCV1; coefficient 4.55; 95% CI: 0.15, 8.96; *p*-value = 0.043) of the DiD coefficients were statistically significant (i.e., no *p*-values < 0.05), but the DiD coefficient was positive with a *p*-value < 0.20 for some vaccines, namely, MCV-1 (coefficient 4.26; 95% CI: −1.30, 9.82; *p*-value = 0.13) and HEPB-3 (coefficient 5.47; 95% CI: −1.45, 12.38; *p*-value = 0.12). The absence of statistical significance of these DiD coefficients could be due to the smaller sample sizes resulting in insufficient statistical power to yield significance.

## Discussion

In this paper, we found preliminary evidence suggesting UHC's potential contribution to safeguarding health service delivery against external shocks. Using the DiD methodology to

**Table 4. Adjusted DiD analysis results of countries with high UHC index values (UHC SCI ≥80) vs. all other countries (UHC SCI <80) in immunization coverage by type of vaccine in pre-pandemic period (2010–2019) compared with pandemic period (2020).**

| Vaccine type | DiD[a] coefficient* | 95% Confidence Interval | *p*-value |
|---|---|---|---|
| All vaccines | 2.7 | (0.75, 4.65) | 0.007 |
| BCG[b] | −2.19 | (−13.94, 9.57) | 0.716 |
| DTP-1[c] | 2.46 | (−1.41, 6.32) | 0.213 |
| DTP-3[d] | 2.36 | (−2.74, 7.47) | 0.364 |
| HEPB3[e] | 5.02 | (−1.11, 11.14) | 0.108 |
| HEPBB[f] | −0.23 | (−22.42, 21.97) | 0.984 |
| HIB-3[g] | 1.74 | (−4.27, 7.76) | 0.57 |
| MCV1[h] | 4.05 | (−1.01, 9.10) | 0.117 |
| MCV-2[i] | 3.77 | (−3.39, 10.93) | 0.303 |
| PCV-3[j] | −2.00 | (−11.41, 7.40) | 0.676 |
| POL-3[k] | 3.13 | (−1.77, 8.02) | 0.211 |
| ROTAC[l] | 0.72 | (−11.38, 12.82) | 0.907 |
| RCV1[m] | 4.55 | (0.15, 8.96) | 0.043 |

*All analyses are adjusted for calendar year, pandemic preparedness, country income group, geographic region, and vaccine types (results full adjusted analyses and unadjusted analyses are available in the Supporting information (Tables A–AA in S2 File).

[a]Difference-in-difference.

[b]Bacille Calmette–Guérin.

[c]Diphtheria, tetanus toxoid, and pertussis containing vaccine—first dose.

[d]Diphtheria, tetanus toxoid, and pertussis containing vaccine—third dose.

[e]Hepatitis B vaccine—third dose.

[f]Hepatitis B vaccine—birth dose.

[g]*Haemophilus influenzae* type B containing vaccine.

[h]Measles containing vaccine—first dose.

[i]Measles containing vaccine—third dose.

[j]Pneumococcal conjugate vaccine—third dose.

[k]Polio containing vaccine—third dose.

[l]Rotavirus vaccine—second or third dose.

[m]Rubella containing vaccine—first dose.

measure the differences in immunization coverage before and during the first year of the COVID-19 pandemic, we observed significantly smaller decline in childhood immunization coverage among countries with greater progress toward UHC.

Our findings suggest that countries with greater progress toward UHC were able to mitigate the decline in childhood immunization coverage during the pandemic year of 2020, even after adjusting for vaccine type and countries' income level, geographic region, and health emergency preparedness and response capacity. This finding may provide potential support to our hypothesis that greater progress toward UHC safeguard countries' ability to provide essential health services, as measured by the proxy of childhood immunizations, during external shocks.

With our robustness checks, we observed that the association between UHC index and change in immunization coverage was only observed among countries with strong UHC performance (UHC SCI 2019 ≥80). Although the same significant impact was not observed when the analysis was performed on each individual vaccine, this lack of statistical significance is likely due to insufficient statistical power stemming from the inherently small sample size of

countries in the world. Nonetheless, to the best of our knowledge, our study is one of the first attempts to quantify the possible contribution of UHC in safeguarding health system performance during a public health crisis. Our findings not only add to the current literature showing the tangible benefits of UHC on health and health systems, thus strengthening the basis of policy dialogue and advocacy to promote UHC, but also underscores the relevance and importance of future research on the protective effects of UHC during public health crises.

There were 2 obstacles to overcome in this analysis. First, studies relying on cross-sectional summary health statistics have shown that countries' progress toward UHC might be irrelevant to COVID-19-related health outcomes [9]. However, summary health statistics may not be appropriate in analyses seeking to examine the complicated mechanisms underpinning countries' pandemic preparedness and response capacity, and more advanced methodologies are needed to test for causality. Second, establishing empirical evidence of the pandemic's impact on countries' service delivery is currently difficult due to the innate delay in the release of pertinent national statistics. Most annual global health statistics require at least a year before their release; the delay will likely be longer for the 2020 data due to the pandemic. Timely evidence, however, is critically important to inform policies aimed at improving countries' health system resilience both now and before the next public health crisis strikes. For this reason, we used currently available data on routine childhood immunization as a starting point and plan to follow up in the future with more complete data on other types of essential health services. Even though our data source for routine childhood immunization is widely used for research exploring country-level immunization coverage [32–34], we acknowledge uncertainty in the data's estimates [17,35]. These uncertainties include country-level estimates not accounting for pockets of low immunization coverage within countries, varying reporting mechanisms and data quality across countries, and differing methods used in the model- and consultation-based adjustments. Even with these limitations, we believe the data source was optimal for testing our hypothesis, which concerns immunization coverage trends rather than point estimates.

We believe that the study can be further improved in 3 ways. First, our findings may still be subject to unobserved confounding factors. Due to data unavailability, we were not able to include several potential confounders as covariates in our analysis, such as countries' average number of healthcare workers per capita, the size of the catchment population for each type of vaccine, the resources allocated for immunization activities, and countries' vaccine availability. Additionally, for simplification, we assumed that all covariates included in our analysis were time-fixed within our analytic timeframe (2010 to 2020) after observing no major change in any of the variables included in our analysis. However, the inclusion of more confounders, including time-varying covariates, may improve future results.

Second, once the immunization coverage data for subsequent years becomes available, the same hypothesis can be tested using a comparative interrupted time series analysis. In this analysis, the one-time change in post-pandemic immunization coverage attributed to UHC's protective effect, which is analogous to the DiD coefficient in our study, can be quantified by the interaction term between the indicator variables defining the pre-/post-periods and the treatment and control groups based on the UHC SCI index. The subsequent year-over-year change in post-pandemic immunization coverage attributed to UHC's protective effect can be quantified by introducing a 3-way interaction term between the prepost variable, the treatment variable, and the year variable centered to the introduction of the shock so that 2020 will be coded with a value of 1, 2021 with a value of 2, and so forth. Using this approach, we will be able to not only quantify the protective effect of UHC in the face of an external shock, but also observe the effect of UHC on the speed of health system recovery in subsequent years. Toward

this end, we aim to monitor the release of data so that we can further evaluate the role of UHC in supporting countries to better respond to—and recover from—public health crises.

Using public health indicators other than immunization coverage to serve as a proxy measure of essential health service delivery can further strengthen our findings. We believe that the HIV continuum of care indicators, such as CD4+ count and viral load, or other indicators related to maternal and newborn health could be robust candidates. Further, the use of key health indicators, such as neonatal, under 5, or maternal mortality rates, would enable us to quantify the role of UHC in safeguarding population health in times of crisis rather than just during peacetime, which has been demonstrated by many prior studies [2–6]. These analyses cannot be performed in the foreseeable future due to the dearth of post-pandemic data available for these indicators. However, once the data becomes available, we strongly believe that replicating this analysis using alternative indicators can validate and reinforce our findings.

Countries' health system resilience against public health emergencies has been studied almost exclusively under the framework of global health security (GHS), with no role of UHC discussed [9]. This clear separation of investigations has precluded the opportunity to examine the potential contribution of UHC in strengthening health system resilience against external shocks; GHS and UHC policies should likely complement each other [9]. In view of the several major outbreaks over the past decade, it is important to understand the synergistic impact of investments in GHS and UHC on countries' health system resilience against external shocks [36,37]. Toward this end, our study provides preliminary evidence on the role of UHC in supporting countries' ability to deliver essential health services in the face of external shocks like the COVID-19 pandemic and sets the stage for future research in this area. Further, our findings strongly suggest that policymakers should continue to advocate for policies aimed at achieving UHC in coming years.

## Supporting information

**S1 Text. References to the datasets used for the analysis.**
(DOCX)

**S1 File. Contains additional descriptive statistics (Tables A-C and Figs A and B).** Paragraph A. Data source, compiled dataset, link to the repository. Table A. Complete list of 195 countries included in the dataset with their UHC SCI 2019 in alphabetical. Fig A. Change in overall vaccine coverage (A) and vaccine specific coverage (B) globally by UHC Service Coverage Index 2019 (UHC SCI <50 vs. UHC SCI ≥50) between 1997 and 2020. Table B. Results of ordinary-square linear regression analysis to assess the parallel pre-trend assumption before the COVID-19 pandemic. Fig B. Histogram of the distribution of UHC SCI 2019 with the cutoff value of 80 for the treatment vs. control group marked with dotted line. Table C. Summary characteristics of the countries included in the analysis ($N$ = 195) based on the cutoff value of 50 for the UHC Service Coverage Index 2019.
(DOCX)

**S2 File. Contains additional results of the regression analysis (Tables A-AA).** Table A. Regression analysis results of countries with high UHC index values (UHC SCI ≥80) vs. all other countries (UHC SCI <80) in childhood immunization coverage in pre-pandemic period (2010–2019) compared with pandemic period (2020)—Base model with no DiD interaction term (Unadjusted). Table B. Difference-in-difference regression analysis of bacille Calmette–Guérin (BCG) coverage after COVID-19 pandemic by UHC SCI 2019 (≥80 vs. the rest)—Unadjusted. Table C. Difference-in-difference regression analysis of bacille Calmette–Guérin (BCG) coverage after COVID-19 pandemic by UHC SCI 2019 (≥80 vs. the rest)—Adjusted

for calendar year, pandemic preparedness, country income group, and geographic region. Table D. Difference-in-difference regression analysis of the first dose of diphtheria and tetanus toxoid and pertussis containing vaccine (DTP1) coverage after COVID-19 pandemic by UHC SCI 2019 (≥80 vs. the rest)—Unadjusted. Table E. Difference-in-difference regression analysis of the first dose of diphtheria and tetanus toxoid and pertussis containing vaccine (DTP1) coverage after COVID-19 pandemic by UHC SCI 2019 (≥80 vs. the rest)—Adjusted for calendar year, pandemic preparedness, country income group, and geographic region. Table F. Difference-in-difference regression analysis of the third dose of diphtheria and tetanus toxoid and pertussis containing vaccine (DTP3) coverage after COVID-19 pandemic by UHC SCI 2019 (≥80 vs. the rest)—Unadjusted. Table G. Difference-in-difference regression analysis of the third dose of diphtheria and tetanus toxoid and pertussis containing vaccine (DTP3) coverage after COVID-19 pandemic by UHC SCI 2019 (≥80 vs. the rest)—Adjusted for calendar year, pandemic preparedness, country income group, and geographic region. Table H. Difference-in-difference regression analysis of the third dose of hepatitis B containing vaccine (HEPB3) coverage after COVID-19 pandemic by UHC SCI 2019 (≥80 vs. the rest)—Unadjusted. Table I. Difference-in-difference regression analysis of the third dose of hepatitis B containing vaccine (HEPB3) coverage after COVID-19 pandemic by UHC SCI 2019 (≥80 vs. the rest)—Adjusted for calendar year, pandemic preparedness, country income group, and geographic region. Table J. Difference-in-difference regression analysis of the birth dose of hepatitis B containing vaccine (HEPBB) coverage after COVID-19 pandemic by UHC SCI 2019 (≥80 vs. the rest)—Unadjusted. Table K. Difference-in-difference regression analysis of the birth dose of hepatitis B containing vaccine (HEPBB) coverage after COVID-19 pandemic by UHC SCI 2019 (≥80 vs. the rest)—Adjusted for calendar year, pandemic preparedness, country income group, and geographic region. Table L. Difference-in-difference regression analysis of the third dose of *Haemophilius influenza* B containing vaccine (HIB3) coverage after COVID-19 pandemic by UHC SCI 2019 (≥80 vs. the rest)—Unadjusted. Table M. Difference-in-difference regression analysis of the third dose of *Haemophilius influenza* B containing vaccine (HIB3) coverage after COVID-19 pandemic by UHC SCI 2019 (≥80 vs. the rest)—Adjusted for calendar year, pandemic preparedness, country income group, and geographic region. Table N. Difference-in-difference regression analysis of the first dose of measles containing vaccine (MCV1) coverage after COVID-19 pandemic by UHC SCI 2019 (≥80 vs. the rest)—Unadjusted. Table O. Difference-in-difference regression analysis of the first dose of measles containing vaccine (MCV1) coverage after COVID-19 pandemic by UHC SCI 2019 (≥80 vs. the rest)—Adjusted for calendar year, pandemic preparedness, country income group, and geographic region. Table P. Difference-in-difference regression analysis of the second dose of measles containing vaccine (MCV2) after COVID-19 pandemic by UHC SCI 2019 (≥80 vs. the rest)—Unadjusted. Table Q. Difference-in-difference regression analysis of the second dose of measles containing vaccine (MCV2) after COVID-19 pandemic by UHC SCI 2019 (≥80 vs. the rest)—Adjusted for calendar year, pandemic preparedness, country income group, and geographic region. Table R. Difference-in-difference regression analysis of the third dose of pneumococcal conjugate vaccine (PCV3) coverage after COVID-19 pandemic by UHC SCI 2019 (≥80 vs. the rest)—Unadjusted. Table S. Difference-in-difference regression analysis of the third dose of pneumococcal conjugate vaccine (PCV3) coverage after COVID-19 pandemic by UHC SCI 2019 (≥80 vs. the rest)—Adjusted for calendar year, pandemic preparedness, country income group, and geographic region. Table T. Difference-in-difference regression analysis of the third dose of polio containing vaccine (POL3) coverage after COVID-19 pandemic by UHC SCI 2019 (≥80 vs. the rest)—Unadjusted. Table U. Difference-in-difference regression analysis of the third dose of polio containing vaccine (POL3) coverage after COVID-19 pandemic by UHC SCI 2019 (≥80 vs. the rest)—Adjusted for calendar year,

pandemic preparedness, country income group, and geographic region. Table V. Difference-in-difference regression analysis of the second or third dose of rotavirus vaccine (ROTAC) coverage after COVID-19 pandemic by UHC SCI 2019 (≥80 vs. the rest)—Unadjusted. Table W. Difference-in-difference regression analysis of the second or third dose of rotavirus vaccine (ROTAC) coverage after COVID-19 pandemic by UHC SCI 2019 (≥80 vs. the rest)—Adjusted for calendar year, pandemic preparedness, country income group, and geographic region. Table X. Difference-in-difference regression analysis of the first dose of rubella containing vaccine (RCV1) coverage after COVID-19 pandemic by UHC SCI 2019 (≥80 vs. the rest)—Unadjusted. Table Y. Difference-in-difference regression analysis of the first dose of rubella containing vaccine (RCV1) coverage after COVID-19 pandemic by UHC SCI 2019 (≥80 vs. the rest)—Adjusted for calendar year, pandemic preparedness, country income group, and geographic region. Table Z. Difference-in-difference regression analysis of overall immunization coverage after COVID-19 pandemic by UHC SCI 2019 (<50 vs. the rest)—Unadjusted. Table AA. Difference-in-difference regression analysis of overall immunization coverage after COVID-19 pandemic by UHC SCI 2019 (<50 vs. the rest)—Adjusted for calendar year, pandemic preparedness, country income group, geographic region, and vaccine types.
(DOCX)

**S3 File. Contains further sensitivity analyses as suggested by reviewers (Tables A-E, Fig A).** Fig A. Adjusted difference-in-difference coefficient from the analysis replicated with a range of cutoff values threshold (50–80) for UHC SCI 2019. Table A. Adjusted difference-in-difference coefficient from the analysis replicated with a range of cutoff values threshold (50–80) for UHC SCI 2019. Table B. Difference-in-difference regression analysis of overall immunization coverage after COVID-19 pandemic by UHC SCI 2019 (> = 80 vs. the rest) from 2015 to 2020—Unadjusted. Table C. Difference-in-difference regression analysis of overall immunization coverage after COVID-19 pandemic by UHC SCI 2019 (> = 80 vs. the rest) from 2015 to 2020—Adjusted for calendar year, pandemic preparedness, country income group, geographic region, and vaccine types. Table D. Difference-in-difference regression analysis of overall immunization coverage after COVID-19 pandemic by UHC SCI 2019 (<50 vs. the rest) from 2015 to 2020—Unadjusted. Table E. Difference-in-difference regression analysis of overall immunization coverage after COVID-19 pandemic by UHC SCI 2019 (<50 vs. the rest) from 2015 to 2020—Adjusted for calendar year, pandemic preparedness, country income group, geographic region, and vaccine types.
(DOCX)

**S1 STROBE Checklist.** Table A. STROBE Statement—Checklist of items that should be included in reports of cross-sectional studies.
(DOCX)

## Author Contributions

**Conceptualization:** Sooyoung Kim, Yesim Tozan.

**Data curation:** Sooyoung Kim.

**Formal analysis:** Sooyoung Kim, Tyler Y. Headley.

**Methodology:** Sooyoung Kim, Tyler Y. Headley, Yesim Tozan.

**Resources:** Yesim Tozan.

**Supervision:** Yesim Tozan.

**Validation:** Tyler Y. Headley, Yesim Tozan.

**Visualization:** Sooyoung Kim.

**Writing – original draft:** Sooyoung Kim.

**Writing – review & editing:** Tyler Y. Headley, Yesim Tozan.

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
