## [Editor Report · Decision Letter 0]

25 Jan 2022

Dear Dr Tozan, 

Thank you for submitting your manuscript entitled "The Role of Universal Healthcare Coverage in Safeguarding Health Service Delivery in Times of Public Health Crises: A Difference-in-Difference Analysis of Childhood Immunization Coverage Before and During the COVID-19 Pandemic" for consideration by PLOS Medicine.

Your manuscript has now been evaluated by the PLOS Medicine editorial staff and I am writing to let you know that we would like to send your submission out for external peer review.

Please re-submit your manuscript within two working days, i.e. by Jan 27 2022 11:59PM.

Kind regards,

Caitlin Moyer, Ph.D.

Associate Editor

PLOS Medicine

---

## [Decision Letter · Decision Letter 1]

18 Apr 2022

Dear Dr. Tozan,

Thank you very much for submitting your manuscript "The Role of Universal Healthcare Coverage in Safeguarding Health Service Delivery in Times of Public Health Crises: A Difference-in-Difference Analysis of Childhood Immunization Coverage Before and During the COVID-19 Pandemic" (PMEDICINE-D-22-00227R1) for consideration at PLOS Medicine. 

Your paper was evaluated by a senior editor and discussed among all the editors here. It was also discussed with an academic editor with relevant expertise, and sent to three independent reviewers, including a statistical reviewer. The reviews are appended at the bottom of this email and any accompanying reviewer attachments can be seen via the link below:

[LINK]

In light of these reviews, I am afraid that we will not be able to accept the manuscript for publication in the journal in its current form, but we would like to consider a revised version that addresses the reviewers' and editors' comments. Obviously we cannot make any decision about publication until we have seen the revised manuscript and your response, and we plan to seek re-review by one or more of the reviewers. 

We expect to receive your revised manuscript by May 09 2022 11:59PM. Please email us (plosmedicine@plos.org) if you have any questions or concerns.

We look forward to receiving your revised manuscript. 

Sincerely,

Caitlin Moyer, Ph.D.

Associate Editor

PLOS Medicine

plosmedicine.org

From the academic editor: From the academic editor: The biggest concern is that the UHC index is highly correlated with national income (and health system spend)--so we may be actually assessing whether rich countries maintained immunization rather than whether UHC per se had an impact. To assess the latter, the authors could do an additional analysis of low-middle and low income countries only. Or split analysis into high income, middle income, and low-income. In any case, as written, I am not sure we can conclude much about UHC as the protective factor.

Other editorial comments:

1. Title: Please revise your title according to PLOS Medicine's style.

-Your title must be nondeclarative and not a question.

-It should begin with main concept if possible.

-"Effect of" should be used only if causality can be inferred, i.e., for an RCT. Please place the study design ("A randomized controlled trial," "A retrospective study," "A modelling study," etc.) in the subtitle (ie, after a colon).

-Please use “sentence case capitalization” for the title. Please capitalize the first word of the subtitle. For example: “Universal healthcare coverage and childhood immunization coverage before and during the COVID-19 pandemic: A difference-in-difference analysis”

2. Data availability statement: The Data Availability Statement (DAS) requires revision. Please update the data availability statement to indicate the where the full data sets and analysis code may be accessed.

For each data source used in your study:

3. Throughout: Please avoid referring to “effect of universal healthcare coverage on childhood immunization coverage” and similar language that strongly implies causality. Please soften language to allow for the possibility of alternate explanations - identified associations may potentially provide support for a causal relationship.

4. Abstract: Background: The final sentence should clearly state the study question.

5. Abstract: Methods and Findings: Please ensure that all numbers presented in the abstract are present and identical to numbers presented in the main manuscript text. Please include a brief mention of the inclusion criteria for the 180 countries, and please explicitly describe the main outcome measures. Please briefly describe how countries’ income group, geographical region, and pandemic preparedness were categorized/assessed. Please describe the threshold of UHC SCI >/=80) to establish the treatment and control groups, in terms of what this metric represents, and the rationale for selecting it.

6. Abstract: Methods and Findings: Please quantify the main results with 95% CIs and p values. Please refer to “statistically non-significant” findings. Please mention the important variables that are adjusted for in the analyses.

7. Abstract: Methods and Findings: In the last sentence of the Abstract Methods and Findings section, please describe the main limitation(s) of the study's methodology.

8. Abstract: Conclusions: * Please address the study implications without overreaching what can be concluded from the data; the phrase "In this study, we observed ..." may be useful. * Please interpret the study based on the results presented in the abstract, emphasizing what is new without overstating your conclusions. Please remove the “Funding” statement from the abstract.

9. Author summary: At this stage, we ask that you include a short, non-technical Author Summary of your research to make findings accessible to a wide audience that includes both scientists and non-scientists. The Author Summary should immediately follow the Abstract in your revised manuscript. This text is subject to editorial change and should be distinct from the scientific abstract. Please see our author guidelines for more information: https://journals.plos.org/plosmedicine/s/revising-your-manuscript#loc-author-summary

10. In-text references: Please use numbers within square brackets to refer to references within the main text. Please place brackets prior to sentence punctuation. Where multiple references are indicated, please do not include spaces within brackets.

11. Main text: Please include line numbers running continuously throughout with the revised version.

12. Introduction: Please provide more rationale and importance of the examination of childhood immunization rates specifically as representative of health service delivery, in general (for example, the public health implications of disruption in delivery of essential childhood vaccinations).

13. Method: In the manuscript text, please indicate: (1) the specific hypotheses you intended to test, (2) the analytical methods by which you planned to test them, (3) the analyses you actually performed, and (4) when reported analyses differ from those that were planned, transparent explanations for differences that affect the reliability of the study's results. If a reported analysis was performed based on an interesting but unanticipated pattern in the data, please be clear that the analysis was data-driven.

14. Methods: Did your study have a prospective protocol or analysis plan? Please state this (either way) early in the Methods section.

15. Methods: Please ensure that the study is reported according to the STROBE guideline, and include the completed STROBE checklist as Supporting Information. Please add the following statement, or similar, to the Methods: "This study is reported as per the Strengthening the Reporting of Observational Studies in Epidemiology (STROBE) guideline (S1 Checklist)."

16. Methods: Please provide additional detail for the UHC Service Coverage Index and 2019 Global Health Security Index (GHSI) in terms of what they represent and how they are incorporated into the analyses. Please also mention how geographical location was assigned/categorized.

17. Methods: Page 6-7: “We hypothesized that greater progress towards UHC…” The use of “greater progress” in the text seems to imply changes in UHC SCI over time, rather than UHC SCI index at a single time point (2019). Please clarify.

18. Methods: Page 8: “For the DiD analysis, we leveraged the COVID-19 pandemic to introduce a “pre-post” variable wherein the years prior to 2020 were defined as “pre” (0) and the pandemic year of 2020 was defined as “post” (1).” Please specify the years.

19. Methods: Please specify the significance level used (eg, P<0.05, two-sided) and the statistical test used to derive a p value. When a p value is given, please specify the statistical test used to determine it.

20. Methods: Ethical approval: Please provide the name(s) of the institutional review board(s) that provided or waived ethical approval for the study.

21. Results: Please clarify “SD” in the text, at first use.

22. Results: Please quantify all results from the analyses with 95% CIs and p values. Where adjusted analyses are presented, please explicitly mention the covariates adjusted for, and please also provide the results of unadjusted analyses.

23. Results: Page 12: “Using the DiD model, we found that countries a high UHC index (UHC SCI 2019 80) prevented a 2.93% reduction in overall childhood immunization coverage during the pandemic year of 2020 (p value= 0.01).” Please temper the use of language implying causality throughout, and please describe this association more clearly, avoiding interpretations such as “prevented” in the sentence. We suggest moving “This finding supports our hypothesis that greater progress towards UHC safeguard countries’ ability to provide essential health services, as measured by the proxy of childhood immunizations, during external shocks.” to the Discussion.

24. Discussion: Please present and organize the Discussion as follows: a short, clear summary of the article's findings; what the study adds to existing research and where and why the results may differ from previous research; strengths and limitations of the study; implications and next steps for research, clinical practice, and/or public policy; one-paragraph conclusion.

25. References: Please use the "Vancouver" style for reference formatting, and see our website for other reference guidelines https://journals.plos.org/plosmedicine/s/submission-guidelines#loc-references

26. Figures and Tables: Please ensure each figure and table (including supporting information tables) has a title and descriptive legend. Please define all abbreviations used in the figure/table within the legend.

27. Table 2 and Table 3: Please include the covariates adjusted for in the legend, and please also present results from unadjusted analyses. Please present 95% CIs in addition to p values. When a p value is given, please specify the statistical test used to determine it.

28. S1 Table: “Full dataset used for this analysis can be found here: LINK” Please provide the link to access the full data set, with a reference. Please update the data availability statement section of the manuscript submission system accordingly.

29. Table S2-1: Please specify the reference groups. Please mention the variables adjusted for in the analyses. Please also provide the 95% CIs, and results of unadjusted analyses.

30. Table S2-2 through S2-14: Please mention the variables adjusted for in the analyses. Please also provide the 95% CIs, and results of unadjusted analyses. Please refer to p values as p<0.001 where applicable.

31. Tables: We suggest including a table showing the characteristics of the countries included, summarizing UHC SCI (with thresholds of 80 and 50), income group, geographical location, GHSI.

Comments from the reviewers:

Reviewer #1: This paper provides a clearly written account of an effort to understand the relationship between universal healthcare coverage and declines in immunization coverage during the pandemic. Overall, the topic is important and the logic of the paper is sound. However, some details, which would be helpful to the reader, are omitted (I've outlined these below).

1. Abstract: This sentence is unclear: "The DiD estimators indicated that countries with a high UHC index (UHC SCI380) prevented a 2.93% reduction in childhood immunization coverage during the pandemic year of 2020 as compared to the control group (p-value=0.01)." I think my confusion comes from the mixing of causal and descriptive language. Did having high UHC prevent a 2.93% reduction or was the reduction among countries with high UHC 2.93% less than the reduction among countries with low UHC?

2. The paper models immunization coverage (which is a proportion) as a function of covariates and DiD variables using linear regression. Given that immunization coverage for some vaccines is near 1 in some groups, linear regression may not be appropriate (that is, it would be easy to obtain predictive values above 1). Some justification for the use of a linear model should be included here. 

3. Did the model perform any type of weighting for population?

4. Page 9, last few sentences: how were covariates included in the model (e.g., linear terms, splines, indicator variables?) and how were functional forms chosen?

5. Page 9, last sentence: how was pandemic preparedness operationalized? Specifically, was UHC part of this definition?

6. Page 12, lines 2-4: was the -2.94% computed from the model without the interaction term? This analysis should be described in the methods section. Moreover, I would change "statistically significant and negative" to "lower", or simply use the second part of this sentence that is easier to understand.

7. How was the summary immunization coverage metric computed (both overall and for individual vaccines? Some details should be included in the Methods. (I see that the text references the WHO/UNICEF Joint Estimates. It would suffice here to give a rough overview of what the immunization coverage metric is and how to interpret it). 

8. The paper used the UHC SCI estimates from 2019. Is there any evidence that the pandemic affected UHC? If so, what are the implications for the results?

9. How were specific vaccines that are not typically administered in one or more countries handled? (e.g., BCG in the US)

Reviewer #2: The authors describe an analysis of the influence of universal health coverage (UHC) on the level of disruption to childhood immunisation due to the COVID-19 pandemic. They used a difference-in-difference approach to examine the relative change for each country given their achieved UHC. They found a higher UHC had a protective effect for routine immunisation services in the pandemic. The paper was generally very well written, made some strong points and I enjoyed reading it.

I have some comments on improvements:

The threshold for the UHC cutoff feels a little arbitrary and I think a small amount of work would help justify this (even with the sensitivity analysis at UHC=50). For example, Figure S1-2 would suggest a cutoff of 75 would split the distributions into two more distinct groups. You could perform a Kolmogorov-Smirnov test on a range of cutoff values to find the most distinct split. In this way, the threshold itself may also be informative.

It would be worth discussing the other indicators used in UHC to highlight the value added by this analysis and to illustrate that high UHC does not necessarily only depend on high immunisation rates. Otherwise, the argument may appear circular.

Whilst vaccination coverage may be easily accessible it is subject to large uncertainties in reporting, denominator and effective impact. Coverage clustering leads to zero dose children this while high numbers of doses are delivered there may still be pockets of low immunisation status which will influence the perceived benefits. Similarly, coverage estimates are based on population estimates that are themselves subject to substantial uncertainty. Finally, reporting mechanisms vary substantially between countries. It would be good to discuss and acknowledge these uncertainties.

Over the years 2010-2019 most countries saw an improvement in achieved coverage- in line with this, the reduction seen for 2020 may appear more stark compared to the end of the time period rather than the average. Would it be feasible to compare a shorter initial time period to capture this variation ie comparing 2015-2019 to the pandemic year.

Can you comment on the code availability. The link in the document does not appear to load.

Reviewer #3: This study explores if the Covid 19 pandemic impact in 2020 on immunization coverage was less in 180 countries with stronger UHC. It compares the coverage trends of 14 childhood vaccines between 2010 and 2020 between countries with higher UHC service coverage index score compared to low score (cut off 80 SCI). The comparison was compared using calendar year, income group, geographical region and preparedness index using global health security index (GHSI). 

The topic is relevant and timely. It adds evidence that stronger health systems are more resilient and that investments in UHC is important in advance of next pandemic

Suggest minor revisions for publication in PLOS Medicine. 

General questions are: 

1. Why was the analysis controlled for by pandemic preparedness? In the IHR the index includes coverage. Could it be that preparedness could be well functioning in a country but health systems performing badly? I wonder if by controlling for it, the very thing you want to study also equals out because GHSI and SCI are linked. Table 3 shows no significance of any individual vaccine in the regression analysis. The explanation in the text instead says it is because of insufficient statistical power. Can you explain? 

2. The conclusion in the abstract to " strongly suggest policymakers to achieve UHC even during a public health crisis". Would suggest "achieve UHC in the coming years". The study did not study the possibility to achieve UHC during the pandemic.

3. Data is used from the end of 2020, but we know that the situation of impact on essential immunization was very dynamic during all of 2020, where Africa was extraordinary resilient compared to other Regions. Would it not have been good in include data on the epidemiologic situation or lockdown measures, which was very impactful reasons for impact on immunization services? 

4. How was the "average coverage rate " calculated? Was it an average of all 14 vaccines by country? 

Minor comments in addition: 

1. In table 1, why do you now mention the countries with less than 80 but more than 50 UHC SCI?

2. HPV is not included in the Fig 1B. Was it used in the analysis? HPV coverage in 2020 was mainly affected by lockdowns and school closure which is less directly related to UHC. 

3. I find it a bit odd to describe in the discussion what equation to use in the next paper. Consider to avoid using the equation and explain plainly what is suggested.

[LINK]

---

## [Decision Letter · Decision Letter 2]

22 Jun 2022

Dear Dr. Tozan,

Thank you very much for re-submitting your manuscript "Universal Healthcare Coverage and Health Service Delivery in Times of Public Health Crises: A Difference-in-Difference Study of Childhood Immunization Coverage Before and During the COVID-19 Pandemic" (PMEDICINE-D-22-00227R2) for review by PLOS Medicine.

I have discussed the paper with my colleagues and the academic editor and it was also seen again by one of the reviewers. I am pleased to say that provided the remaining editorial and production issues are dealt with we are planning to accept the paper for publication in the journal.

[LINK]

We look forward to receiving the revised manuscript by Jun 29 2022 11:59PM.   

Sincerely,

Caitlin Moyer, Ph.D.

Associate Editor 

PLOS Medicine

plosmedicine.org

Requests from Editors:

1. Title: Please indicate the number of countries in the title.

2. Short title: Please specify the “COVID-19 pandemic” or similar.

3. Data availability statement: Thank you for including the github dataset. Please provide information on how to access the WHO/UNICEF Joint Estimates of National Immunization Coverage, The IHME UHC Service Coverage Index, World Bank Classification for income groups, and the GHSI for 2019. (The references provided in “S1 Dataset used for the analysis” might be useful).

4. Abstract: Line 24-25: We suggest “...quantified the relationship between UHC and childhood immunization coverage before and during the COVID-19 pandemic.”

5. Abstract: Please combine the Methods and Results sections into one section, “Methods and findings”.

6. Abstract: Line 27: It may be helpful to mention that only 12 vaccines were considered in the final analyses.

7. Abstract: Line 35: Here, and throughout the manuscript, please replace “less reduction” with “smaller reduction” or similar.

8. Abstract: Line 37: Rather than “control group” it would be helpful to explain these are the countries with UHC less than 80.

9. Abstract: Lines 35-38: For both the analysis with UHC SCI > or = 80 and 50, please give the numbers of countries that fell above and below these thresholds.

10. Abstract: Line 42: Please replace “less declines” with “smaller declines” or similar.

11. Abstract: Line 43: Please reword this to avoid drawing conclusions about causality. “...may potentially provide support for the importance of UHC in building health system resilience.” or similar.

12. Author summary: Line 62-64: Please use “smaller decline” here. Please also describe the control group.

13. Author summary: Line 65-66: Please remove this point.

14. Introduction: Line 101: Here and throughout, please remove spaces from within brackets [9,10].

15. Introduction: Line 128-131: Please revise throughout to remove emphasis on causal implications. “This goal of the study was to examine the relationship between UHC and health service delivery during the COVID-19 pandemic shock.” or similar may be useful.

16. Methods: Hypothesis: Please move the hypothesis to the end of the Introduction.

17. Methods: Line 140: We suggest “These data include…” or “This dataset includes…” or similar.

18. Methods: Line 143: Please remove spaces from within reference brackets.

19. Methods Line 164: The text indicates that compiled data is presented in paragraph 1 of S1 file. Please make it clear that this paragraph contains information to access the full dataset as well as the information about the original data.

20. Methods: Line 185-187: “Lastly, even in resource-limited countries with poor health service provisioning and lower UHC index values, childhood immunization programs are prioritized and perform relatively well compared to other service areas.” Please provide a reference for this sentence.

21. Methods: Line 218-226: Please provide a brief explanation for why these particular vaccine/dose combinations were selected as indicators (e.g. why the third PCV dose, but the birth dose of Hep B?) if possible.

22. Methods: Data analysis: Please specify the significance level used (eg, P<0.05, two-sided) and the statistical tests used to derive a p value.

23. Results: Line 277: Here and throughout, it may be helpful to consistently refer to UHC index less than or greater/equal to 80 (similar to lines 273-276) rather than switch to “treatment” and “control” groups.

24. Results: Line 296-299: Although this is implied, it would be helpful to please also present here the Pre-post coefficient for UHC SCI greater than equal to 80 and less than 80, with 95% CIs and p values, to support that those countries with an index of 80 and greater experienced no significant change while the decline was significant in countries with an index of less than 80.

25. Results: Line 303: Please use “not significant” here.

26. Results: Line 306-310: “Subsequently, when we repeated the same analysis using the sliding threshold of UHC SCI between 50 and 80 with step increments of five, we observed a significant association between UHC and a differential drop in immunization coverage during the pandemic from the threshold value of 65 and above (Supporting Information Figure S3-1).” Please quantify these results in Figure S3-1, with the difference-in-difference coefficients, 95% CIs, and p values. Please indicate in the legend the tests used to determine significance.

27. Results: Line 340: Please use “absence of statistical significance” here.

28. Discussion: Line 365-368: We suggest rewording to: “Using the DiD methodology to measure the differences in immunization coverage before and during the first year of the COVID-19 pandemic, we observed significantly less decline in childhood immunization coverage among countries with greater progress toward UHC.” or similar.

29. Discussion: Line 369: We suggest revising to: “Our findings suggest that countries with greater progress towards UHC were able to mitigate the decline in childhood immunization coverage…”

30. Discussion: Line 375: Please revise to: “With our robustness checks, we observed that the association between UHC index and change in immunization coverage was only observed among countries with strong UHC performance…”

31. Discussion: Line 379-381: Please revise to: “Nonetheless, to the best of our knowledge, our study is one of the first attempts to quantify the possible contribution of UHC in safeguarding health system performance during a public health crisis.”

32. Discussion: Please be sure that the Discussion is organized as follows: a short, clear summary of the article's findings; what the study adds to existing research and where and why the results may differ from previous research; strengths and limitations of the study; implications and next steps for research, clinical practice, and/or public policy; one-paragraph conclusion.

33. References: Please use the "Vancouver" style for reference formatting, and see our website for other reference guidelines https://journals.plos.org/plosmedicine/s/submission-guidelines#loc-references

-Please pay particular attention to the journal title abbreviations (e.g. The Lancet should be Lancet in Reference 2 and throughout).

-Please provide complete details for reference 12, reference 13, reference 15, reference 18.

-Please change the journal title abbreviation to BMJ Glob Health for reference 17.

-Please change the journal title abbreviation to Lancet Infect Dis for reference 27.

-Please change the journal title abbreviation to Lancet Glob Health for reference 29.

-Please change the journal title abbreviation to BMC Int Health Hum Rights for reference 34.

-Please change the journal title abbreviation to PLOS Glob Public Health for reference 35.

34. Table 1: It might be helpful to also include a section of the table listing those countries that fall between UHC SCI 80 and 50, as these would be the ones to “switch” from control to treatment groups between the analyses.

35. Figure 1: Please define all abbreviations used in the legend.

36. Supporting information Table S1-3: We suggest moving the table of summary characteristics for the countries to the main text of the manuscript.

37. Supporting information Tables S3-4: Please make it clear in the title and/or legend that for these analyses you are replicating using a pre period of 2015-2019.

38. Supporting Information 4: STROBE Checklist: We suggest including this as a separate document.

Comments from Reviewers:

Reviewer #1: The authors have addressed my previous comments.

[LINK]

---

## [Editor Report · Decision Letter 3]

29 Jun 2022

Dear Dr Tozan, 

On behalf of my colleagues and the Academic Editor, Margaret Kruk, I am pleased to inform you that we have agreed to publish your manuscript "Universal Healthcare Coverage and Health Service Delivery in Times of Public Health Crises: A Difference-in-Difference Study of Childhood Immunization Coverage from 180 Countries Before and During the COVID-19 Pandemic" (PMEDICINE-D-22-00227R3) in PLOS Medicine.

Please also address the following editorial points:

-Title: Please capitalize only the first word of the title, and first word of the subtitle (after the colon). Please shorten the title to: “Universal healthcare coverage and health service delivery before and during the COVID-19 pandemic: A difference-in-difference study of childhood immunization coverage from 180 countries”

-Introduction: Line 85: Please use “particularly for people with limited financial resources” or similar.

-Introduction: Line 95: Please revise to: “among disadvantaged socioeconomic groups” or similar.

-Methods: Line 153-154: Please mention the fiscal year used for the World Bank classification of income groups.

-Methods: Line 193-194: Please revise to: “...countries that made less progress towards UHC…”

-Table 3: Please correct two typos for the p values (p=0) indicated for Southeast Asia under “WHO region” in the table.

-Figure 2: Please change the y axis label to: “Drop in childhood immunizaiton coverage (%) during 2020” or similar, and please change the legend accordingly. Please indicate in the legend if the bars represent 95% CIs, or other.

-Reference 15: Please change the journal title abbreviation to: Pediatr Res.

-Reference 28: Please change the journal title abbreviation to: Int Health.

-Reference 31: Please revise the author, correct the weblink and revise format. Please also correct this in file S1.

PRESS

Sincerely, 

Caitlin Moyer, Ph.D. 

Associate Editor 

PLOS Medicine